

# Organic and inorganic nitrogen amendments suppress decomposition of biodegradable plastic mulch films

Sreejata Bandopadhyay[1*], Marie English[1], Marife B. Anunciado[1], Mallari Starrett[1], Jialin Hu[1], José E Liquet y González[1], Douglas G. Hayes[1], Sean M. Schaeffer[1], Jennifer M. DeBruyn[1]

[1]Department of Biosystems Engineering and Soil Science, University of Tennessee, Knoxville, Tennessee, United States

*Present address: Department of Microbiology and Molecular Genetics, Michigan State University, East Lansing, Michigan, United States

*Correspondence to* Jennifer DeBruyn (jdebruyn@utk.edu)

**Abstract** Biodegradable mulch films (BDMs) are a sustainable and promising alternative to non-biodegradable polyethylene mulches used in crop production systems. Nitrogen amendments in the form of fertilizers are used by growers to enhance soil and plant-available nutrients, however, there is limited research on how these additions impact biodegradation of BDMs tilled into soils. A four-month soil microcosm study was used to investigate the effects of inorganic (ammonium nitrate) and organic (urea and amino acids) nitrogen application on biodegradable mulch decomposition. We investigated the response of soil bacterial, fungal and ammonia-oxidizing microbial abundance along with soil nitrogen pools and enzyme activities. Microcosms were comprised of soils from two diverse climates (Knoxville, TN, USA and Mount Vernon, WA, USA) and BioAgri, a biodegradable mulch film made of Mater-Bi®; a bioplastic raw material containing starch and poly(butylene adipate-co-terephthalate) (PBAT). Both organic and inorganic nitrogen amendments inhibited mulch decomposition, soil bacterial abundances and enzyme activities. The greatest inhibition of mulch biodegradation in TN soils was observed with urea amendment where biodegradation was reduced by about 6% compared to the no-nitrogen control. In WA soils, all nitrogen amendments suppressed biodegradation by about 1% compared to the no-nitrogen control. Ammonia monooxygenase *amoA* gene abundances were increased in TN soils in all treatments, but reduced for all treatments in WA soils. However, a significantly higher nitrate and lower ammonium concentration was seen for all nitrogen treatments compared to no-nitrogen controls in both TN and WA. This study suggests that addition of nitrogen, particularly inorganic amendments, could negatively affect mulch decomposition but that mulch decomposition does not negatively affect soil nitrification activity.

**Short Summary** We added organic and inorganic nitrogen amendments to two soil types in a laboratory incubation study in order to understand how that would impact biodegradable plastic mulch (BDM) decomposition. We found that nitrogen amendments, particularly urea and inorganic nitrogen, suppressed BDM degradation in both soil types. However, we found limited impact of BDM addition on soil nitrification, suggesting that overall microbial processes were not compromised due to the addition of BDMs.



## 1 Introduction

Plastic mulch films provide several key benefits to crop production systems. In addition to conserving soil moisture and moderating soil temperatures, mulches suppress weed growth and limit contamination of crops by soil (Hayes et al., 2019; Kasirajan and Ngouajio, 2012; Serrano-Ruiz et al., 2021). The development of a conducive microclimate enhances crop yields (Miles et al., 2012; Moreno and Moreno, 2008). Multiple crops are grown using mulch films such as wheat (Chakraborty et al., 2008; Li et al., 2005), cotton (Chen et al., 2014) and vegetables such as okra, squash

and bell peppers (Mahadeen, 2014; Filipović et al., 2016). Most of the mulch films used in agriculture are made of polyethylene (PE) which is poorly biodegradable and requires disposal after the growing season, typically ending up in landfills (Hayes et al., 2012). Fragments of PE mulches may remain in the soil and cause potential harm to soil and aquatic ecosystems (Bandopadhyay et al., 2018). Biodegradable mulch films (BDMs) are a sustainable alternative to PE films because they are made of biodegradable polymers such as starch, polyesters and lignin. BDMs are meant to

be tilled into soils where microbes convert them into carbon dioxide, water and microbial biomass.

One concern that growers have regarding BDMs is the unpredictable breakdown in the field. Both above-soil and in-soil degradation rates differ among BDMs and climates. There is no standard currently available pertaining to BDM degradation in field conditions, however there are several which certify biodegradability of BDMs. ASTM D6400 (2012) and an International Organization for Standardization (ISO) equivalent (Dentzman, 2019) tests biodegradation

under industrial composting conditions and is the most commonly cited standard in reference to BDMs. The European Committee for Standardization (CEN) released the European Standard EN 17033 in January 2018, which was the first standard to certify plastic mulch films as "biodegradable" in ambient soil (Cen, 2018), and uses laboratory tests mimicking field soil conditions. Both ASTM D6400 and EN 17033 address inherent biodegradability and ecotoxicity. The EN 17033 standard requires 90% biodegradation in two years using ASTM D5988 test, and the ASTM D6400

standard requires 90% biodegradation within 6 months by the ASTM D5338 test. Of course, even BDMs meeting the standards in laboratory tests may have variable in-field degradation performance: Climate, crop, weathering and soil management can all influence biodegradation rates and complete biodegradation is not always achieved (Anunciado et al., 2021; Hayes, 2018).

Because of variable degradation rates of BDMs in the field, understanding controls on BDM biodegradation is critical

to support use of BDMs within the farming community. The possible accumulation of undegraded BDM fragments in soil over time could impact soil infiltration rates further affecting water uptake by plants and microbes. Even though BDMs are a small carbon input compared to extant soil carbon and crop residues (Ding et al., 2021), BDM fragments in soil may affect ecosystem functions over the long term. Short-term studies of four years demonstrate that BDMs have comparable effects to PE films and do not alter soil quality and soil biological functions significantly in that time

period (Sintim et al., 2019; Sintim et al., 2021). However, effective ways to facilitate mulch decomposition in the soil are warranted in a way that would not adversely affect soil health.

Combining plastic mulching with fertilizer application has been known to increase crop yield almost two fold (Li et al., 2003; Lamont, 2005). Fertilizers have long been used in the field by growers to improve soil functioning and crop




yields by providing essential nutrients to plants. Common inorganic nitrogen fertilizers include ammonium nitrate,
urea, ammonium sulphate and diammonium phosphate and organic fertilizers commonly include amendments such as
manure and straw. Even though addition of organic fertilizers can increase bacterial to fungal ratio in soil (Marschner
et al., 2003) and rapidly increase $CO_2$ evolution (Ajwa and Tabatabai, 1994), their nutrients are slowly released and
hence often not immediately available to plants. Inorganic fertilizers act rapidly but can accumulate toxic
concentration of salts in soils and vegetables if over-applied (Santamaria, 2006). When fertilizers are used in soils,
these nutrient pulses could shift soil C:N ratios, causing enhanced or reduced decomposition of organic material in
the soil (Marschner et al., 2003).

It has long been recognized that soil mineral elements, particularly nitrogen, play a critical role in constraining the
rate of organic matter decomposition. Increased organic matter decomposition due to nitrogen deposition is well
documented in the literature (Hobbie, 2000; Hobbie and Vitousek, 2000; Hunt et al., 1988; Knorr et al., 2005; Boxman
et al., 1995; Vitousek et al., 1997). Nitrogen additions are also known to accelerate decomposition of light soil carbon
fractions (Neff et al., 2002). Even though most studies report a positive effect of nitrogen addition on decomposition
of plant litter and organic matter, a few studies also point towards a negative effect of nitrogen on microbial activity
and organic matter decomposition (Fog, 1988; Fang et al., 2007; Janssens et al., 2010). Thus, mixed results are
encountered when looking at the effects of nitrogen on stability and turnover of soil carbon pools.

Some environmental factors that affect biodegradation rates of BDMs are considered to be 1) environmental conditions
such as temperature, moisture, pH, and oxygen availability; and 2) physicochemical properties of the polymeric
materials (Kijchavengkul et al., 2008). However, information regarding the impacts of soil-specific factors such as
nitrogen content on BDM degradation processes is sparse. It has been proposed that nitrogen-limited microorganisms
may not produce esterases that degrade BDMs because hydrolysis of polyesters does not increase nitrogen availability
for the organism (Sander, 2019). One study showed better correlation of the degree of degradation of biodegradable
plastics with the total nitrogen content of soil than with the total carbon content (Hoshino et al., 2001). Another study
by Thompson et al. (2019) evaluated the effects of urea and other biodegradation amendments on poly-lactic acid
(PLA) mulch degradation in microcosms, but the impacts of such amendments on mulch decomposition and microbial
respirations were reported to be inconsistent across the different PLA mulch types throughout the study.

The overall objective of our study was to better understand how nitrogen amendments influence biodegradation of
BDMs in soil. Our specific objectives were to determine if: 1) addition of nitrogen amendments in the form of organic
and inorganic treatments stimulates the decomposition of biodegradable mulch films, and 2) addition/degradation of
BDMs has an impact on soil nitrification activity. Since the polymers used in BDMs are composed of C, O, and H,
microbes need to acquire nitrogen from the surrounding soil for growth. Therefore, microbes may experience nitrogen
limitation in the soil or on the surfaces of rapidly depolymerizing C-rich polymers. Thus, we hypothesized that
nitrogen amendments would enhance mulch decomposition and nitrification by alleviating N-limitation compared to
treatments with no added nitrogen. We also hypothesized that biodegradation of mulch material would have minimal
effect on soil nitrification processes based on our prior observations that BDMs had minimal impact on soil microbial



communities and biogeochemical cycling in field soils (Sintim et al., 2019; Sintim et al., 2021; Bandopadhyay et al.,
105   2020).

## 2   Materials and Methods

### 2.1 Site description and soil sampling

Soils for this experiment were collected from two locations: East Tennessee Research and Education Center (ETREC),
University of Tennessee, Knoxville, Tennessee, USA and the Northwestern Washington Research & Extension Center
(NWREC), Washington State University, Mount Vernon, Washington, USA. The soil at Knoxville is a sandy loam
(59.9% sand, 23.5% silt, and 16.6% clay), classified as a fine kaolinitic thermic Typic Paleudults. The soil at Mount
Vernon is a silt loam (14.2% sand, 69.8% silt, and 16% clay), classified as a fine-silty mixed nonacid mesic Typic
Fluvaquents. Soil and weather characteristics are included in Table 1.

**Table 1: Soil and weather characteristics in Knoxville, TN and Mount Vernon, WA (Sintim et al., 2019).**

| | Knoxville, TN | Mt. Vernon, WA |
|---|---|---|
| Soil type | Typic Hapludult | Typic Fluvaquent |
| Texture | 59.9% sand | 14.2% sand |
| | 23.5% silt | 69.8% silt |
| | 16.6% clay | 16% clay |
| pH (Sintim et al., 2019) | 6.03 | 6.24 |
| CEC ($cmol_c$ $kg^{-1}$) (Sintim et al., 2019) | 7.23 | 9.19 |
| Nitrate-N (mg $kg^{-1}$) | 20.9 | 4.49 |
| P (mg $kg^{-1}$) | 72.6 | 77.4 |
| Organic matter (%) (Sintim et al., 2019) | 1.43 | 2.36 |
| Mean annual precipitation (mm) | 1355 | 831 |
| Mean daily annual temperature (℃) | 14.1 | 10.5 |
| %Carbon (%C) | 0.746 | 1.228 |
| %Nitrogen (%N) | 0.079 | 0.115 |
| C:N ratio | 9.43 | 10.68 |



Soil samples were obtained from control (unmulched) plots from a vegetable cropping field experiment comparing different mulch treatments (experimental details can be found in (Sintim et al., 2021). Thirty soil cores were collected

using soil augers (0 to 10 cm depth) from the four replicated unmulched plots in TN and WA in July 2018. Soils were collected into pre-sterilized buckets and any roots or debris were picked out. Soils from one plot were composited by homogenizing by hand and then put into Ziplock® bags. These were stored in a cooler with ice packs for transport to the laboratory. Field-moist soil was then passed through a 2–mm sieve to remove any remaining coarse rock and plant material. Soils from the four no-mulch plots at each location were then mixed together to form a homogenized soil

sample. These soils were kept at room temperature for 7 days at 25ºC.

### 2.2 Biodegradable mulch used for incubation study

BioAgri, a starch-PBAT (poly (butylene adipate-co-terepthalate)) blend BDM made by BioBag® USA was used in this study. Field-weathered BioAgri plastic mulch samples (hereafter, "plastic") were collected at the end of the 2015 growing season at the TN field site after three months of exposure to the environment. Film thickness and surface area

density of the weathered mulch was $0.078 \pm 0.0188$ mm and $25.17 \pm 4.578$ gm$^{-2}$ respectively (Hayes et al., 2017). Mulch pieces were cleaned using a dry, clean brush to dust off soil by gentle dabbing prior to use. Plastic pieces were cut into 2 cm * 2 cm pieces and 250 mg plastic weighed in aluminum tins to be added to respective microcosms. About 25 to 30 mulch pieces were added per jar in the study.

### 2.3 Laboratory incubation experimental design

After incubation at room temperature, 100 g of field moist soil was weighed into 473 ml glass canning jars and adjusted to 50% water holding capacity (WHC) by adding deionized water. After adjusting WHC, jars containing soils were pre-incubated in the dark at 25ºC for an additional 5 days. The experiments consisted of two treatment factors, plastic addition (0 mg and 250 mg), and nitrogen amendments in the form of urea ($CH_4N_2O$, from Fisher Chemical, purity 99.7%), ammonium nitrate ($NH_4NO_3$, from Fisher Chemical, purity 99.9%), amino acids (as a complete supplemental

medium (CSM) powder from Sunshine Science Products), and a no-nitrogen control. 790 mg of the CSM powder contained the following amounts of each amino acids: adenine hemisulfate (10 mg), L-arginine (50 mg/L), L-aspartic acid (80 mg), L-histidine hydrochloride monohydrate (20 mg), L-isoleucine (50 mg), L-leucine (100 mg), L-lysine hydrochloride (50 mg), L-methionine (20 mg), L-phenylalanine (50 mg), L-threonine (100 mg), L-tryptophan (50 mg), L-tyrosine (50 mg), L-valine (140 mg) and uracil (20 mg). At the start of the incubation experiment, jars

containing soil were taken out of incubator and nitrogen amendments were added at a rate of 50 mg N kg$^{-1}$ soil on an N equivalent basis, similar to field application rates in TN. Soils were taken out of pre-incubated jars and put in a large weigh boat where the soil was mixed with the respective amendments using a spatula. 1 ml of 0.2 M solution of urea and ammonium nitrate, and 1 ml of 31.6 g L$^{-1}$ CSM, each dissolved in deionized (DI) water, was added to the respective soils and 1 ml of DI water was added to soils designated as no-nitrogen control. All treatments were done in triplicate.

After mixing of amendments, soil and field-weathered plastic pieces (pre-cut into 2 cm * 2 cm squares) were arranged alternately in the jars in a way such that two layers of plastic remained buried within the soils, ensuring that all the plastic was covered by soil. Incubation was carried out in the dark at 30ºC for 16 weeks after amendment application. A total of 48 experimental units were set up in a full factorial design: 2 soil types x 2 plastic treatments x 4 nitrogen



treatments x 3 replicates x 1 time point at 16 weeks. Time zero samples were collected on the day of start of experiment before nitrogen application.

**2.4 Soil analyses before incubation**

For all t = 0 measurements, 10 additional jars were set up with soils adjusted to 50% WHC (five from TN, five from WA). These were also pre-incubated along with all the experimental units. On the day of start of the experiment (t = 0, when N amendments were added), 1 ml of DI water was added to one of the jars to account for change in moisture

after adding nitrogen amendment and soils from that jar were stored in the -80ºC freezer for DNA extractions and enzyme assays to be done later. Gravimetric moisture content was determined for these soils which was used for dry soil weight calculations. $K_2SO_4$ extractions were completed on soil samples at t = 0 with added nitrogen amendments to account for any differences in carbon and nitrogen pools which might result due to addition of N. For this, 5 g of soils from three of the five replicate jars were weighed out and urea, ammonium nitrate and amino acids (CSM) were

added at a rate equivalent to the amount added to 100 g soil in microcosms. 20 ml of 0.5M $K_2SO_4$ was added to these jars and extractions were completed. All samples were shaken in an incubator shaker at 160 rpm for four hours, after which these were filtered through a vacuum manifold using 1 μm glass microfiber filters and stored at -20ºC. These extracts were used for nitrate and ammonium assays.

Soils (40 mg) were collected before and after the incubation, dried at 60 °C and ground and subsequently analyzed

for total %C and total %N using a Costech Elemental Analyzer (Costech, EA ECS4010) at the Stable Isotope Laboratory in the Department of Earth and Planetary Sciences at the University of Tennessee. L-Glutamic Acid and acetanilide were used as calibration standards for percent C and N.

**2.5 Gas chromatography and microcosm maintenance**

Headspace gas samples were collected in 12 ml gas vials starting from the day of start of experiment. Gas sampling

was done every two to three days for the first two weeks after which it was reduced to once per week sampling until day 119 (~16 weeks). For the first day of gas sampling, after addition of N amendments, jars were vented and then first gas samples were taken using a 30 ml syringe. 15 ml of gas was used to over pressurize 12 ml vials (pre-evacuated to 200 mTorr). Needles were cleaned out between samples to avoid cross contamination. For subsequent gas sampling days, jars were vented for a minute after gas sample collection to avoid excessive gas build-up in jars. Preliminary

results from a trial run indicated the need to dilute gas samples. Thus, before reading gas samples, all vials were diluted in highly pure $N_2$-gas about 13-fold. Diluted gas samples were then read in a gas chromatograph (Shimadzu GC-2014, Shimadzu, Japan) following the methods described in Hu et al. (Hu et al., 2021b). Majority of the dilutions were prepared by one person to avoid human variation introduced with different techniques of dilution.

Microcosms were opened every sampling day to vent and during that time jars were weighed and kept at a constant

moisture. Jars weights taken at t = 0 were used as reference and the weights were kept constant by adding DI water as needed. Any unburied plastic pieces were re-buried. Gas samples were also taken after venting of jars was complete to make sure that the venting method was yielding similar gas concentrations in all jars. GC data confirmed that the venting was consistently done throughout the experiment.



### 2.6 Soil analyses after incubation

**2.6.1 Soil physicochemical analyses**

At the end of the experiment, soil samples from each microcosm were sieved through a 2 mm sieve to separate plastics and soil samples. Soils from each jar was weighed out for pH, electrical conductivity (EC), gravimetric soil moisture, $K_2SO_4$ extractions, nitrate assays, ammonium assays, DNA extractions and enzyme assays.

Gravimetric soil moisture was measured on the soils collected at t = 16 weeks by weighing ~3 g of soil from the jars

and then drying them in an oven (Heratherm OGS100, Thermo Scientific) for 48 h at 105℃. The dry weight was noted after 48 h and then the gravimetric soil moisture was calculated using Equation 1.

$$\% \text{ gravimteric soil moisture} = \left( \frac{\text{Mass}_{\text{wet soil}} - \text{Mass}_{\text{dry soil}}}{\text{Mass}_{\text{dry soil}}} \right) * 100 \qquad \text{(Equation 1)}$$

where:

$\text{Mass}_{\text{wet soil}}$ = Mass of wet soil (g)

$\text{Mass}_{\text{dry soil}}$ = Mass of dry soil (g)

pH and electrical conductivity (EC) were measured from soils collected at t = 16 weeks by preparing soil slurries (5 g soil: 10 mL deionized water). pH was measured using a benchtop pH meter (SevenCompact™ Benchtop pH meter, Mettler Toledo) whereas EC was measured using a handheld multiparameter meter (Orion Star A329, Thermo Scientific) with each probe being calibrated prior to use. EC was measured immediately after vortexing the slurry, and

pH was measured after approximately 15 min, allowing the fine particles to settle.

Soils were processed to determine %C and %N values similarly as the pre-incubation samples. Untreated soil samples from final and initial time points (without added nitrogen amendments and mulch) were used to calculate changes in %C, %N and C:N ratios. The relative change in C:N ratio was calculated as per Equation 2 (Schmidt and Gleixner, 2005).

$$\text{Relative change in C: N ratio} = 100 * \frac{\text{C:N}_{\text{after incubation}} - \text{C:N}_{\text{before incubation}}}{\text{C:N}_{\text{before incubation}}} \quad \text{(Equation 2)}$$

### 2.6.2 Quantitative PCR

Extraction of DNA from soil samples was completed using the MoBio™ PowerLyzer™ Power Soil DNA isolation kit (now Qiagen™ PowerSoil kit) per manufacturer's instructions. 0.25 grams of soil were used for the extractions,

and the DNA obtained after the final elution step (60 µl) was stored at $-20^0$C until further analyses. The extracted DNA was used to quantify bacterial, fungal and nitrifier gene abundances in the soil collected both at t = 0 and t = 16 weeks.

As a proxy for bacterial and fungal abundances, 16S rRNA (bacteria) and ITS (fungi) gene copy numbers were quantified from soil DNA samples using Femto™ Bacterial DNA quantification kit (Zymo Research) and Femto™



Fungal DNA quantification kit (Zymo Research) following the manufacturer's protocol. DNA extracts were diluted
      1:10 prior to quantification. All samples were analyzed in triplicate. No-template negative controls were included in
      each run. Bacterial and fungal DNA standards were provided in the kit and used to calculate to copy numbers. qPCR
      reactions were performed in a CFX Connect Real-Time PCR Detection System (BioRad). qPCR efficiencies averaged
      around 79% for bacterial and fungal assays. Standard curves had R squared values ranging from 0.98 to 0.99.

Quantitative PCR of ammonia-oxidizing bacteria (AOB) *amoA* genes was performed using the SYBR Green Master
      Mix and a CFX Connect Real-Time PCR Detection System (Bio-Rad laboratories, Hercules, CA, USA) following the
      procedure described in Hu et al. (2021b). Briefly, primers amoA1F and amoA2R (Rotthauwe et al., 1997) were used
      and standard curves were constructed with plasmids containing cloned *amoA* products from environmental DNA.
      qPCR efficiencies averaged around 86%. Standard curves had R squared values ranging from 0.98 to 0.99.

**2.6.3 Soil enzyme assays**

      Fluorescence microplate enzyme assays were conducted to determine soil enzyme activity rates before and after the
      incubation study using the protocol described in Bandopadhyay et al. (2020). Briefly, assays were completed using
      fluorescently labelled substrates targeted for three common carbon and nitrogen cycling enzymes that are known to
      degrade sugar, chitin and cellulose (Bell et al., 2013). The synthetic fluorescent indicators used were 4-
methylumbelliferone (MUB) and 7-amino-4-methylcoumarin (MUC). The enzyme activity was measured as the
      fluorescent dye was released from the substrate by an  enzyme-catalyzed reaction, with higher fluorescence indicating
      more substrate degradation compared to lower fluorescence. The targeted enzymes, their respective fluorescently
      labelled substrates are given in Table S1.

      **2.6.4 Nitrate and ammonium assays**

Nitrate and ammonium concentrations in soil samples before and after 16 weeks incubation were determined using
      colorimetric assays. $K_2SO_4$ extractions were completed after 16 weeks on all 48 soil samples collected from the
      experimental units. 10 g of soil was mixed with 40 ml of 0.5M $K_2SO_4$ and blanks were also extracted using only
      $K_2SO_4$. All samples were shaken in an incubator shaker at 160 rpm for four hours, after which these were filtered
      through a vacuum manifold using 1 µm glass microfiber filters and stored at -20ºC. Nitrate and ammonium were
measured on these extracts. Ammonium in soil extracts was measured according to Rhine et al. (1998). Briefly, 70 µl
      of sample extract and ammonium standards (ammonium sulfate [$(NH_4)_2SO_2$]) were pipetted into respective wells of a
      96-well plate; all samples were analyzed in triplicate. Reagents (50 µl citrate reagent, 50 µl 2-phenylphenol-
      nitroprusside, 25 µl buffered hypochlorite reagent, 50 µl milli-Q water) were allowed to react with the sample by
      shaking for 30s on a table shaker. After incubation at room temperature for 2 hours, absorbance (660 nm) was
measured on a Biotek® Synergy H1 plate reader. Some samples were run again using a 1:1 dilution so that they could
      be read using the standard curve; samples were diluted in 0.5M $K_2SO_4$. Nitrate in soil extracts was measured according
      to (Doane and Horwáth, 2003). 30 µl of sample and nitrate standards (potassium nitrate, $KNO_3$) were added to
      respective wells in 96-well plates; all samples were analyzed in triplicate. The reagent, vanadium (III) chloride solution





(300 µl), was then added to each well and the plate incubated for 5 hours at room temperature. Absorbance (540 nm)
was measured on the plate reader. All samples were diluted 1:10 in 0.5M $K_2SO_4$.

## 2.7 Plastic analysis after incubation

### 2.7.1 Plastic sampling strategy and imaging

Soils were sieved post incubation to separate remaining plastics. Plastics collected from one of the replicates were laid
out on a white paper (12 cm*12 cm) and imaged using a 12-megapixel camera, which was put on a level to stabilize.
Visual percent biodegradation, estimated through the loss of surface area for the mulch film pieces, was determined
using Image-J software (Schneider et al., 2012) using the following steps: the scale was set to a ruler included in the
image, the image was converted to an 8-bit image and the threshold was adjusted to black and white before analyzing
the particles (Cowan et al., 2013). One concern was that some particles were larger than the starting value of 4 cm$^2$
indicating that some plastics were fused together. Therefore, the visual estimates may underestimate the actual amount
of plastic remaining. Plastic mulches recovered from the first replicate microcosms for each treatment were also
imaged on a Zeiss EVO MA 15 scanning electron microscope.

### 2.7.2 Percent biodegradation calculation using $CO_2$ evolution data

Based on the ASTM D5988-18 standard (Astm, 2018) the extent of aerobic biodegradability of BioAgri was
determined by measuring evolved $CO_2$ as a function of time that the plastic was exposed to the soil. The starting
amount of plastic carbon in the soil microcosms was determined to be 1322.5 µg plastic carbon g$^{-1}$ dry soil for TN
jars and 1437.5 µg plastic carbon g$^{-1}$ dry soil for WA jars. Using the $CO_2$ evolution data (µg-C released g$^{-1}$ dry soil),
the amount of plastic carbon released over 16 weeks was calculated. This along with the starting plastic carbon value
was used in Equation 3 to determine percent biodegradation for each treatment after 16 weeks.

$$\%\text{bio}_{\text{theo}} = \left( \frac{M_{CO2-C,\text{mulch}} - M_{CO2-C,\text{no-mulch}}}{M_{C,\text{mulch}}} \right) * 100 \qquad \text{(Equation 3)}$$

where:

$\%\text{bio}_{\text{theo}}$ = Theoretical % biodegradation

$M_{CO2-C, \text{mulch}}$ = $CO_2$-C evolved in mulch treatments (µg C g$^{-1}$ dry soil)

$M_{CO2-C, \text{no-mulch}}$ = $CO_2$-C evolved in treatments without mulch (µg C g$^{-1}$ dry soil)

$M_{C,\text{mulch}}$ = Total amount of C added in the form of mulch C (µg C g$^{-1}$ dry soil)

### 2.7.3 Thermogravimetric analysis (TGA)

TGA was carried out for BioAgri plastics retrieved from microcosms after 16-week incubation and compared to the
starting material (agriculturally-weathered plastic stored at room temperature) following the protocol by Hayes et al.
(2017). Briefly, the test was conducted using a Discovery TGA (TA Instruments, New Castle, DE USA) at a heating





rate of 10 °C min$^{-1}$ from room temperature (25 °C) to 600 °C in an unsealed platinum sample pan under a nitrogen
        atmosphere which underwent dynamic flow.

### 2.7.4 ATR-FTIR spectrometry and Gel permeation chromatography

        Attenuated Total Reflectance (ATR)-FTIR absorbance spectra (4000-600 cm$^{-1}$) were collected to determine
        changes in chemical bonding that resulted from the nitrogen treatments following the protocol described by (Hayes
et al., 2017). Briefly, spectra were taken of the plastics, before and after laboratory incubation using an IRAffinity-1
        spectrometer (Shimadzu Co., Tokyo, Japan) equipped with a single reflection ATR system (MIRacle ATR, PIKE
        Technologies, Madison, WI, USA). The parameters for measurement included a resolution of 2 cm$^{-1}$ and 16 scans
        per spectrum. Data were corrected for the background signal attributable to the ATR crystal. Spectral data were
        normalized by equating the integrated peak area of the entire spectrum to 1.0 (mean normalization). Gel permeation
chromatography was performed as described in Hayes et al. (2017).

### 2.7.5 Statistical analysis

        A Student's t- test was conducted to look at differences in bacterial 16S, fungal ITS and *amoA* gene abundances;
        nitrate and ammonium concentrations and enzyme activity rates at t = 16 weeks from t = 0. Additionally, for statistics
        reported in figures, this study used a two-way ANOVA with plastic and nitrogen amendment as factors to isolate the
effects of nitrogen treatment on all response metrics, separating out the location variable prior to two-way ANOVA
        testing. For mulch component analysis (% starch, % PBAT) through TGA, data were subjected to two-way analysis
        of variance (ANOVA) using N amendment and location as factors. Differences between means were performed using
        Tukey's honestly significant difference (HSD) analysis at significant difference level of α = 0.05. Three-way ANOVA
        values including effects of location, plastic and nitrogen amendment were also conducted (results reported in
Supplemental Information). Analysis was conducted using R version 3.4.0. When the ANOVA indicated significant
        effects (p < 0.05), a post hoc analysis with Tukey's HSD (Honest significance test) was applied to compare the means.
        The experiment design included three replications of each treatment. Data were checked for normality using the
        Shapiro-Wilk test (W > 0.9). Outliers were removed and if normality was not achieved the data was log transformed
        as appropriate. For microbial abundances, nitrate and ammonium concentrations and enzyme activity rates, the starting
value of the soil was subtracted from the measured value after 16 weeks and this change score was used for analyses.

### 3 Results

        To understand if location had a significant effect on the different parameters measured, we conducted three-way
        ANOVA Type III tests (Table S2). We found significant main effects and interaction effects with location for almost
all parameters measured and hence we focused on two way-ANOVA results within TN and WA sites separately (Table
        2). The two-way ANOVAs tested differences between nitrogen amendments and plastic addition levels for the
        different parameters evaluated: bacterial, fungal and *amoA* gene abundances; nitrate and ammonium concentrations;
        and enzyme activities.



### 3.1 Soil physicochemical response

When comparing treatments with and without plastic using two-way ANOVAs, it was observed that ammonium nitrate significantly reduced EC with plastic compared to ammonium nitrate treatment without plastic for TN soils, whereas for WA soils, all treatments had significantly reduced EC with plastic as compared to without plastic. Urea application resulted in significantly greater pH with plastic for both TN and WA soils. When comparing across treatments without added plastic, all nitrogen amendments showed significantly greater EC compared to no-nitrogen control for both TN

and WA soils. On the other hand, all nitrogen amendments showed significantly reduced pH compared to no-nitrogen without plastic for both TN and WA soils. For treatments with added plastic, all nitrogen amendments resulted in significantly elevated EC compared to the treatment without nitrogen in both TN and WA soils. All nitrogen amendments resulted in significantly reduced pH compared to the treatment without nitrogen in TN and WA soils, except for urea treatment with plastic in WA soil, where there was no difference with no- nitrogen treatment with

plastic. Significant interaction effects were seen between nitrogen and plastic for all pH and EC data in TN and WA soils, except for EC measurements in WA soil where main effects for nitrogen and plastic treatments are reported (Fig. 1, Table 2).

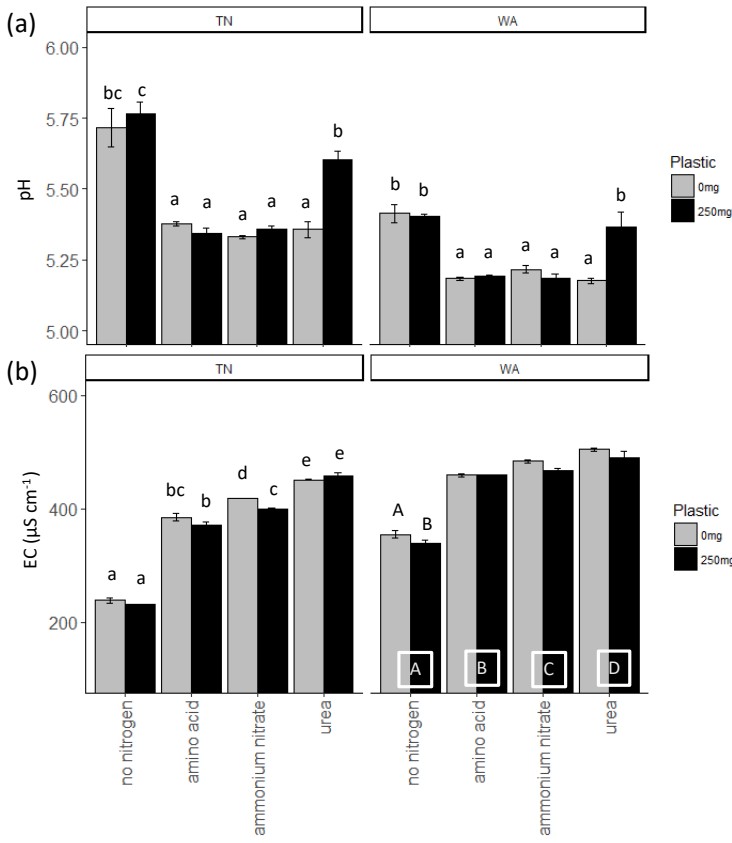



**Fig. 1: (a) pH and (b) electrical conductivity (EC) measurements at t = 16 weeks. Lowercase letters indicate significant interaction effects at α ≤ 0.05. For EC measurements in WA, uppercase letters along x-axis denote a significant main effect of nitrogen amendment at α ≤ 0.05. Uppercase letters on top of the bars in EC plot for WA indicate a significant main effect of plastic at α ≤ 0.05. TN: Tennessee, WA: Washington.**

Soil C:N ratios decreased for nitrogen treatments as well as no-nitrogen controls in TN and WA after incubation (Table S3). When analyzing only the treatments which did not receive any added nitrogen or mulch, we observed a reduction in the %C, %N and C:N values after 16 weeks (comparing final to initial time point, Table S3). The relative change in C:N ratio was calculated for the no-mulch, no-nitrogen controls and revealed a 6% decrease for TN soil and 9% decrease for WA soil.



**Table 2: F values from a two-way ANOVA showing effects of nitrogen and plastic treatment on soil chemical and biological characteristics. Significant differences are in bold; *p < 0.05; **p < 0.01; ***p < 0.001.**

| Location | Factor | pH | EC | $CO_2$-C (µg C $g^{-1}$ dry soil) | Log (bacterial gene copies $g^{-1}$ dry soil) | Log (fungal gene copies $g^{-1}$ dry soil) | Log (*amoA* gene copies $g^{-1}$ dry soil) | $NO_3$ (µg $NO_3$ $g^{-1}$ dry soil) | $NH_4$ (µg $NH_4$ $g^{-1}$ dry soil) | BG (nmol activity $g^{-1}$ dry soil $hr^{-1}$) | CB (nmol activity $g^{-1}$ dry soil $hr^{-1}$) | NAG (nmol activity $g^{-1}$ dry soil $hr^{-1}$) |
|---|---|---|---|---|---|---|---|---|---|---|---|---|
| TN | Nitrogen | 61.26 *** | 557.51 *** | 30.02 *** | 16.11 *** | 10.01 *** | 5.22 * | 174.86 *** | 62702 *** | 7.39 ** | 157.95 *** | 20.15 *** |
|  | Plastic | 9.78 ** | 6.1 * | 304.18 *** | 6.52 * | 0.15 | 1.67 | 5.37 * | 0.08 | 1.98 | 13.52 ** | 14.16 ** |
|  | Nitrogen *Plastic | 6.64 ** | 4.31 * | 12.68 *** | 16.29 *** | 6.33 ** | 7.45 ** | 0.51 | 0.11 | 0.13 | 5.05 * | 2.87 |
| WA | Nitrogen | 57.38 *** | 242.94 *** | 11.76 *** | 1.01 | 5.78 ** | 4.67 * | 27.53 *** | 71573 *** | 2.25 | 3.39 * | 5.33 ** |
|  | Plastic | 8.34 * | 7.86 * | 158.58 *** | 10.46 ** | 5.10 * | 9.10 ** | 0.31 | 1.25 | 0.43 | 0.21 | 0.09 |
|  | Nitrogen *Plastic | 12.08 *** | 1.08 | 1.49 | 3.44 * | 0.74 | 1.21 | 0.03 | 2.25 | 1.77 | 1.29 | 0.92 |

EC: electrical conductivity, BG: β-glucosidase, CB: β-D-cellubiosidase, NAG: N-acetyl β-glucosaminidase




### 3.2 Plastic decomposition in microcosms

Nitrogen addition resulted in similar responses in $CO_2$ evolution ($\mu g$ C $g^{-1}$ dry soil) in both TN and WA soils over the 16-week incubation (Fig. 2a). When analyzing the cumulative $CO_2$-C evolved after 16 weeks using two-way ANOVA, significant interaction effects were seen between nitrogen and plastic in TN, hence, interaction effects are reported (Table 2, Fig. 2b). In WA, interaction effects were not significant, hence, main effects of nitrogen amendment and plastic addition are reported. When plastic was added, cumulative $CO_2$-C respiration was significantly greater for the no-nitrogen control compared to amino acid, ammonium nitrate and urea in TN, and ammonium nitrate and urea in WA (TN no-nitrogen: 230.4 $\mu g$ C $g^{-1}$ dry soil, WA no-nitrogen: 155.06 $\mu g$ C $g^{-1}$ dry soil, Fig. 2b). The amino acid treatment had the least reduction in $CO_2$-C evolution compared to no-nitrogen control over time in both locations, whereas the inorganic amendments urea and ammonium nitrate had the greatest reduction in $CO_2$-C evolution over 16 weeks. Plastic treatments showed significantly greater $CO_2$-C evolution compared to treatments without plastic in both TN and WA indicating biodegradation was occurring. When only nitrogen amendments were added without plastic, $CO_2$-C evolved was significantly higher for amino acid amendment compared to ammonium nitrate in TN whereas in WA, $CO_2$-C was significantly greater for no-nitrogen and amino acid compared to ammonium nitrate and urea. The increased plastic degradation observed with no-nitrogen and amino acid treatments in TN and WA was corroborated visually by images of remaining plastic material in microcosms taken after 16 weeks (Fig. 3, Supplemental Figures S1 and S2) and reflected in maximum theoretical percent biodegradation estimates (Table S4) and visual % degradation via surface area calculation in ImageJ (Fig. S3).

Thermogravimetric analysis (TGA) results were analyzed using thermograms (Fig. 4 a, b) and differential thermograms (the first derivative of the thermograms or DTGs, Fig. 4 c, d) of BioAgri after nitrogen amendment to the soil. Time and temperature are directly proportional to each other, since the heating rate was constant, at 10°C min$^{-1}$. Peaks for the two main heating stages occur at 310°C and 390°C, which represent starch and PBAT, respectively. The relative proportion of mass loss during a heating stage, represented by the area underneath the curve for the DTG peaks, represent the relative mass fractions for the components. The onset, maximum, and final temperatures of the heating stage for starch and PBAT components of BioAgri are contained in Supplemental Table S5. A few nitrogen amendments led to a shift to lower temperatures for the PBAT heating stage: TN-urea, TN-ammonium nitrate and WA-amino acid; most easily observable by the DTGs (Fig. 4 c, d). All nitrogen treatments led to a broadening of the starch heating stage, suggesting starch's microbial utilization. Nitrogen treatment increased the % weight remaining at 600°C; specifically, for ammonium nitrate in TN and amino acid in WA (Supplemental Table S5). Any residual material remaining at 600°C is likely inorganic components of the mulch; and soil particulates adhered to the mulch, and/or the formation of gelled polymer formation due to UV radiation from sunlight which is unlikely to occur in a soil environment. Therefore, % weight remaining at 600°C is expected to increase if biodegradation must occur, due to utilization of the polymeric components as carbon sources for microorganisms. After incubation in TN soil, the % weight remaining at 600°C was highest for ammonium nitrate (33%) and no-nitrogen (33%), followed by urea (28%) and amino acid (20%) treatments. In contrast, for WA % weight remaining was higher for amino acids (37%) and no-nitrogen (23%) amendments. Both ammonium nitrate and urea treatments in WA had comparable % weight remaining, at ~18%.



The fraction of starch and PBAT as components of BioAgri mulch was determined after the 16-week incubation (Fig.
S4) using the thermal degradation process. Comparison of mass fractions of PBAT between soils from TN and WA

did show trends. Starch significantly decreased in nitrogen amendment treatments (p = 0.0344) compared to
agriculturally-weathered control but was not significantly different between two soil types used (p = 0.0542). For both
TN and WA soils, amino acid and ammonium nitrate treatments led to more loss of starch components than other
treatments.

The gel permeation chromatography analysis of molecular weight data represents solely PBAT since PBAT is the

only component of BioAgri that dissolves in chloroform. Loss of weight-averaged molecular weight was observed for
all nitrogen treatments, including no-nitrogen, for both TN and WA soils compared to agriculturally-weathered control
(Supplemental Table S6). The decrease of weight-averaged molecular weight, $M_w$, was not statistically significant (p
= 0.0591) among the nitrogen treatments and between soils (p = 0.4829). Polydispersity index (PDI), however, which
reflects the distribution of molecular weight values for the polymeric strands, increased after nitrogen addition in TN

soil (p = 0.0072) with significant statistical difference among treatments (p = 0.0091). The no-nitrogen controls for
both soils underwent a PDI increase similar to all nitrogen treatments in TN soils, but not for any of the nitrogen
amended soil in WA (Supplemental Table S6).

FTIR data shows changes in the spectra for amino acid and ammonium nitrate treatments, particularly in TN: the
increase of peak intensity of -OH stretching (3700-3000 cm$^{-1}$) and a shift in the carbonyl (C=O) spectral region to

lower wavenumbers (1800-1600 cm$^{-1}$), especially the formation of a band at 1650 cm$^{-1}$ which likely represents COOH
end groups (Fig. 5). Peak assignments for FTIR analysis shown in Supplemental Table S7). These changes on FTIR
spectra reflects hydrolysis occurred on the surface of mulch films when soil was amended with amino acid and
ammonium nitrate substrates, likely a result of biodegradation (Fig. 5). In contrast, the FTIR spectrum for urea
treatment was nearly identical with the spectrum for the control, suggesting that the extent of hydrolysis was relatively

small in the presence of urea.





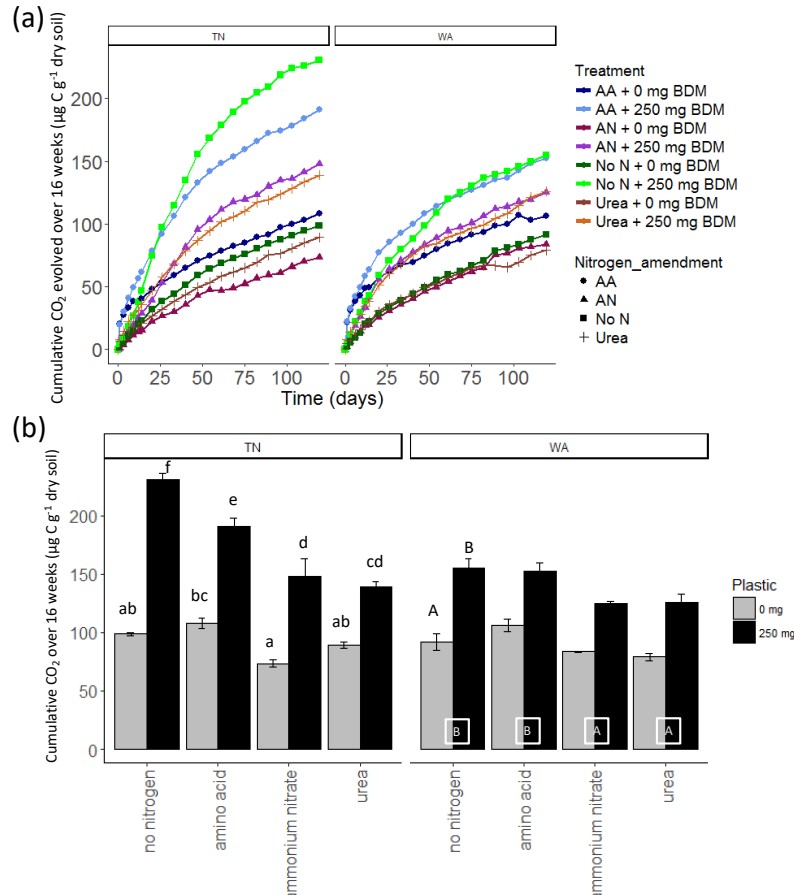

Fig. 2: (a) Cumulative $CO_2$-C released over 16 weeks (119 days) in two soil (TN and WA) microcosms with and without plastic biodegradable plastic mulches (BDMs). AA=amino acids, AN=ammonium nitrate, No N=no nitrogen added. (b) Cumulative $CO_2$-C released over 16 weeks. Each bar represents a mean of 3 replicate microcosms and error bars are standard error. Lowercase letters in TN indicate interaction effects at $\alpha \le 0.05$. In WA, uppercase letters along x-axis indicates a significant main effect of nitrogen treatment at $\alpha \le 0.05$. Uppercase letters on top of the bars in WA indicate a significant main effect of plastic at $\alpha \le 0.05$. TN: Tennessee, WA: Washington.




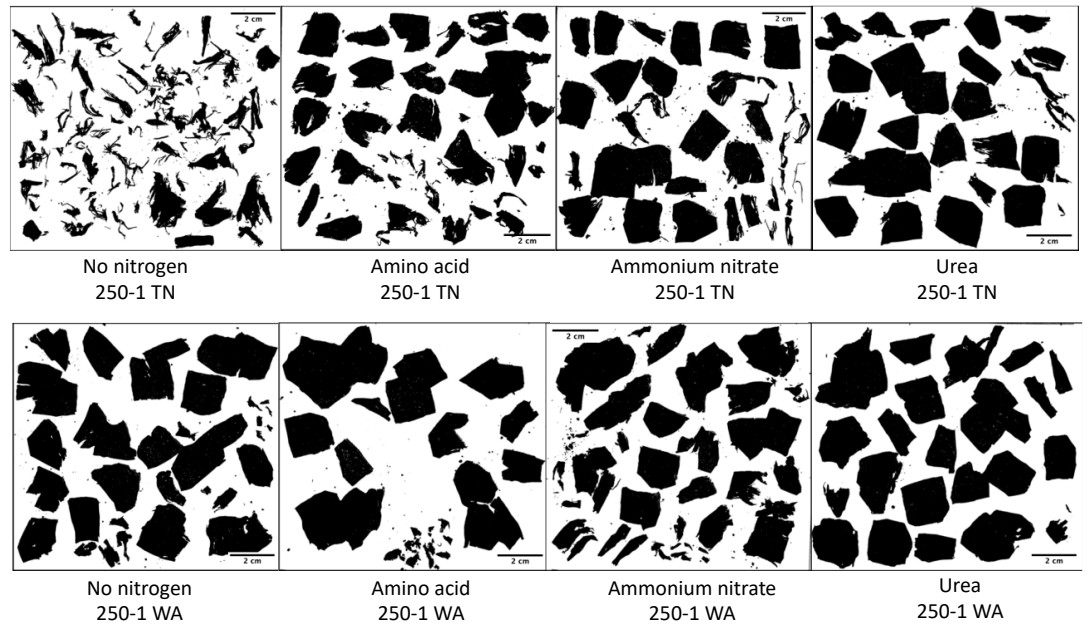


**Fig. 3: Black-white contrast images in ImageJ of plastic pieces after 16 weeks incubation. All images were taken from mulches from the first replicate microcosm for each treatment. TN: Tennessee, WA: Washington.**

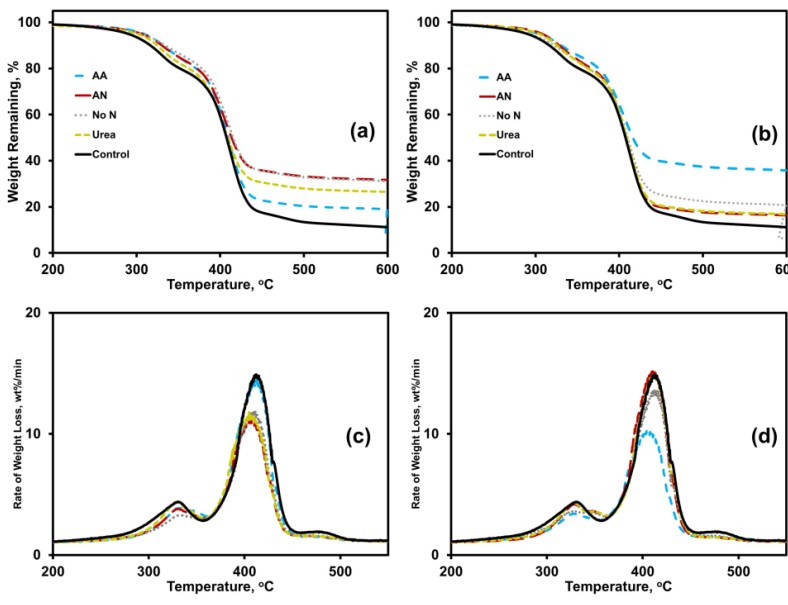





**Fig. 4. Effect of nitrogen amendments on BioAgri plastic as revealed by thermogravimetric analysis (TGA thermograms; a, b) and differential thermograms (DTG; c, d). Plastic were exposed to either TN (a, c) or WA (b, d) soils. AA: Amino acid, AN: Ammonium Nitrate, No N: no nitrogen added. TN: Tennessee, WA: Washington. *Control* refers to agriculturally-weathered BioAgri plastic mulch samples in TN (Hayes et al., 2017).**


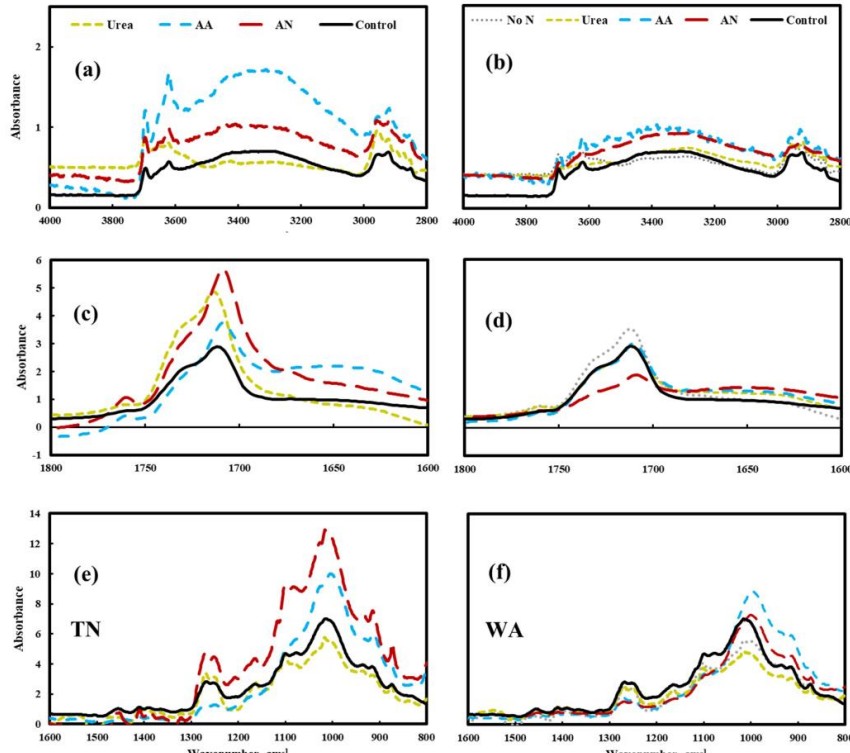


**Fig. 5. Changes in chemical bonds of biodegradable mulch (BioAgri) after biodegradation in nitrogen-amended soil (left panel a, c, e: TN, right panel b, d, f: WA) via FTIR analysis. Mulches retrieved from no-nitrogen (TN) control were too degraded for FTIR analysis. AA: amino acid, AN: ammonium nitrate, No N: no nitrogen added. TN: Tennessee, WA: Washington. *Control* refers to agriculturally-weathered BioAgri plastic mulch samples in TN (Hayes et al., 2017).**


### 3.3 Bacterial and fungal abundances in microcosms

For bacterial abundances, significant interaction effects were observed between nitrogen and plastic factors in TN and WA using two-way ANOVAs, hence, interaction effects are reported (Fig. S5a, Table 2). For fungal abundances, interaction effects were observed with TN but not with WA soils (Fig. S5b, Table 2). When comparing plastic

treatments, it was observed that plastic addition tempered the bacterial reduction effect from ammonium nitrate with treatments without plastic showing significantly reduced bacterial abundance compared to treatments with plastic in



TN. Similarly, in WA, the only significant difference was seen with the no-nitrogen control with plastic addition tempering the bacterial reduction effect. There were no significant differences in fungal abundance between treatments with and without plastic in TN for any of the nitrogen treatments. However, for WA soils, plastic addition caused a

significantly greater fungal abundance compared to treatments without plastic for all nitrogen amendments.

Effects of nitrogen amendments on bacterial and fungal abundances were evaluated for treatments without added plastic (Fig. S5). It was seen that without plastic, ammonium nitrate resulted in significantly reduced bacterial abundance in TN compared to other nitrogen treatments. However, no differences were observed between nitrogen treatments without plastic in WA. Ammonium nitrate without plastic also resulted in significantly reduced fungal

abundances compared to amino acid and urea without plastic in TN, whereas the no-nitrogen control had significantly reduced fungal abundance compared to amino acid treatment. In WA, amino acid treatment without plastic showed significantly greater fungal abundances compared to other treatments without plastic. For treatments with plastic, nitrogen amendments did not cause a significant change in bacterial abundances for TN and WA. Nitrogen treatments with plastic also did not cause a significant change in fungal abundances for TN; however, in WA, amino acid

treatment resulted in a significantly greater fungal abundance compared to other treatments. Overall, for both TN and WA, a reduction in bacterial abundance was observed after 16 weeks. However, there was a significant increase in fungal abundance for both locations over 16 weeks.

### 3.4 Nitrification processes and extracellular enzymatic functions in soil microcosms

For ammonia monooxygenase (*amoA*) gene abundances, two-way ANOVAs revealed significant interaction effect

between nitrogen and plastic in TN while main effects are reported for WA (Table 2). When comparing between plastic treatments, the no-nitrogen control with plastic was shown to cause a significantly higher *amoA* gene abundance compared to no-nitrogen control without plastic in TN (Fig. 6a). In WA, treatments without plastic showed a significantly greater reduction in *amoA* gene abundances when compared to plastic treatments for all nitrogen amendments. Urea without plastic treatments in TN showed significantly greater *amoA* abundance compared to

ammonium nitrate and no-nitrogen without plastic, whereas, in WA, amino acid and urea without plastic showed significantly reduced *amoA* abundances compared to no-nitrogen without plastic. No significant difference in *amoA* gene abundance was observed for nitrogen treatments with plastic in TN. In WA, amino acid and urea with plastic showed significantly reduced *amoA* abundances compared to no-nitrogen with plastic. Overall, trends differed between TN and WA, with increased *amoA* abundance in TN but a decreased abundance in WA after 16 weeks.

Nitrification was increased post nitrogen amendment addition as demonstrated via changes in nitrate and ammonium concentrations after 16 weeks as revealed by two-way ANOVAs (Fig. 6b, c). All added nitrogen treatments resulted in significantly increased nitrate and decreased ammonium concentrations compared to no-nitrogen controls in both TN and WA soils after 16 weeks. Plastic treatments had significantly lower nitrate concentration compared to treatments without plastic in TN. The results were supported by *amoA* gene abundances in TN which were

significantly increased from t = 0 for urea and amino acid treatments with no plastic added, and no-nitrogen and ammonium nitrate with added plastic.



When comparing nitrogen treatments without plastic addition, it was seen that ammonium nitrate caused a significantly greater reduction in BG activity compared to no-nitrogen and amino acid in TN. For CB activity in TN, amino acid and urea had a significant reduction in activity compared to no-nitrogen and ammonium nitrate which caused increases in CB activity after 16 weeks. In WA, urea caused a significantly greater reduction in CB activity compared to ammonium nitrate. Nitrogen amendments without plastic caused a significantly greater reduction in NAG activity in TN compared to no-nitrogen control, whereas in WA, urea and no-nitrogen caused a greater reduction in NAG activity compared to ammonium nitrate (Fig. S6).

When comparing nitrogen treatments with added plastic, it was observed that ammonium nitrate had significantly reduced BG activity compared to no-nitrogen and amino acid in TN after 16 weeks. Amino acid and urea caused significantly greater reductions in CB activity compared to ammonium nitrate in TN, whereas only the no-nitrogen control had an increase in CB activity after 16 weeks. In WA, no-nitrogen, urea and amino acid amendments caused reduced CB activity after 16 weeks, with only ammonium nitrate having a significant increase in CB activity compared to the reduction observed for urea treatment. All nitrogen amendments with plastic had reduced NAG activity compared to no-nitrogen in TN, whereas in WA, no-nitrogen and urea caused significant reductions compared to an increase in NAG activity for ammonium nitrate (Fig. S6).

## 4   Discussion

### 4.1 Nitrogen application suppressed plastic decomposition in both TN and WA soils

The plastics added in the microcosms underwent biodegradation, as evidenced by the increased $CO_2$-C released in microcosms with added plastic, with about 10% and 4% biodegradation in no-nitrogen controls in TN and WA, respectively. The greater mulch decomposition in TN may be explained by a lower C:N ratio compared to WA soils suggesting increased availability of bacterial cellular resources to utilize carbon substrates. The occurrence of biodegradation is also supported by the decrease of molecular weight for PBAT (GPC analysis), the utilization of starch, the anticipated carbon source for the early stage of biodegradation (TGA analysis), and evidence of hydrolysis on the surface of the mulches (FTIR analysis), with the observed changes being greater for TN than for WA. There was some discrepancy between the theoretical biodegradation percentages and the visual estimates reported by imaging of plastics after incubation and calculation of their surface area in ImageJ. In certain treatments, it was observed that the plastics were fused together to form larger particle sizes whereas in no-nitrogen controls the plastics were too degraded to be able to physically recover them from the jars. Hence, it is likely that the visual estimates were underestimating the actual area of plastic remaining post incubation.

Contrary to our hypothesis that nitrogen addition would stimulate plastic decomposition, we instead saw a suppression of plastic decomposition with added nitrogen amendments. Addition of nitrogen resulted in reduction of plastic biodegradation by 6% over 16 weeks for urea treatment in TN; in WA, all the nitrogen treatments showed 1% reduction in biodegradation. The $CO_2$-C results were validated visually, with decreased macroscopic plastic degradation apparent in nitrogen-amendment microcosms. Even though TN and WA soils had different starting soil properties, the



trends for repression in $CO_2$-C were similar between treatments in both locations, indicating that both soils were probably not nitrogen limited to begin with. Studies with forest soils have reported comparable findings where nitrogen deposition was shown to prevent organic matter decomposition when nitrogen was not limiting (Janssens et al., 2010). However, in such a situation it would be expected that the responses to nitrogen addition would be comparable to no-nitrogen controls. The fact that there was a suppression in microbial activity in response to nitrogen amendments in the present study suggests a possible negative effect. Excessive nitrogen deposition in an ecosystem can reach a point of nitrogen saturation which could have severe environmental impacts on soil chemistry and water quality (Aber, 1992). There could also be a decrease in productivity in terrestrial ecosystems through the loss of base cations and decreased phosphorus availability (Jin-Yan and Jing, 2003; Janssens et al., 2010). A study by Thompson et al. (2019) measured biodegradation of different PLA mulches in response to biodegradation amendments such as urea and reported that PLA mulch with composites such as soy particles experienced greater degradation in the presence of urea when compared to other PLA mulches without composites. It could be that the addition of labile carbon to the PLA could enhance biodegradation in response to nitrogen amendments, and that we did not see this affect with our PBAT mulch because it lacked a more labile additive. The decreased mulch biodegradation in the nitrogen addition treatments could also be partly explained by acidification of the soils. Highest mulch biodegradation occurred in the no-nitrogen controls which also had the highest pH. Low pH has been shown to limit carbon use efficiency by increasing fungal abundance (Silva-Sánchez et al., 2019). In agreement with this, we observed an increased fungal abundance for some nitrogen treatments with added mulch compared to no-nitrogen.

The form of nitrogen added dictated the magnitude of the degradation repression especially in TN soils. Addition of amino acids in the form of a complete supplemental media resulted in greater plastic decomposition compared to urea. This may be because in addition to N, amino acids supply labile carbon and sulfur which may have stimulated microbial activity. In the absence of plastic, microbial activity was highest for the amino acid amendment in both TN and WA. Complex organic nitrogen sources have been reported to have a positive effect on organic matter decomposition in other studies, due to carbon source or a vitamin effect rather than a nitrogen induced effect (Fog, 1988). It should also be noted that our study was carried out over a period of four months. A detailed review of various studies reported by Fog (1988) indicates that the repressive effects of nitrogen addition are more apparent in longer term (months) studies, because: 1) nitrogen addition can affect the competition between potent and less potent decomposers, 2) 'ammonia metabolite repression' occurs, where nitrogen blocks production of certain enzymes, especially in fungi, and 3) production of toxic or inhibitory compounds formed by the reaction of amino compounds with polyphenols and other decomposition products. Although it is generally known that addition of nitrogen does not always promote decomposition of organic matter it was not until the research conducted by Keyser et al. (1978) that it was clearly demonstrated how nitrogen can have a negative effect on production of certain enzymes involved in organic matter decomposition in fungi, particularly in Basidiomycetes.

In our study we use the term biodegradation to refer to the theoretical biodegradation of mulch which was computed by subtraction of no-mulch controls and assumes that $CO_2$ produced in the plastic treatments derives from the plastics. We acknowledge that there could be priming effects of added mulch inputs to soil carbon decomposition that




complicates this assumption. Priming, defined as the change in microbial decomposition of soil organic carbon in response to fresh carbon inputs (Bastida et al., 2019), can be considered as extra decomposition of native soil organic matter in a soil receiving an organic amendment. Lower nitrate values in the plastic treatments in TN could be an indicator of microbial assimilation of nitrogen required to support the uptake of plastic carbon by microbes (Bettas Ardisson et al., 2014). In the present study, it is difficult without labeled substrates/mulches to assess how much of the carbon is directly evolved from the plastic vs. enhanced $CO_2$ release from soil organic matter due to the addition of mulch. This limitation is acknowledged, and future experiments with labeled mulches could precisely track these pools of carbon.

There was good correlation between the percent degradation values from the $CO_2$ data and the percent weight remaining from TGA for all the treatments in TN and WA. There is evidence that nitrogen amendment caused depolymerization of PBAT, as shown by the greater decrease of $M_w$ in TN for nitrogen treatments compared to unamended soil (GPC analysis, Table S6) and the greater extent of hydrolysis on the surface, particularly for amino acid and ammonium nitrate treatments in TN (FTIR analysis, Fig 5). Interestingly, specific treatments affected microbial communities' utilization of carbon source: for instance, ammonium nitrate and amino acid amendments and the no-nitrogen control underwent greater utilization of starch by microbes as demonstrated via TGA analysis (Fig. S4).

**4.2 Responses of soil bacterial, fungal and ammonia-oxidizing bacteria to nitrogen and plastic amendments**

An overall decrease in bacterial gene abundance was observed for all treatments after 16 weeks. However, a significantly enriched fungal gene abundance was observed over 16 weeks for all nitrogen treatments with and without plastic in TN and WA. Increased fungal abundances in all nitrogen treatments with added plastic, coupled with reduced plastic decomposition observed in only nitrogen added treatments points towards a possibility of ammonia metabolite repression in response to the added nitrogen. Such a repression has been demonstrated with Basidiomycetes where enzymes degrading organic matter, such as lignin, are produced when the fungus has switched from primary to secondary metabolism. This switch is induced only when the supplies of available nitrogen, such as ammonium, have been exhausted (Fog, 1988). Several studies point towards the important role of fungi in the degradation of biodegradable mulch with evidence of enrichment of *Aspergillus* and *Penicillium* (Koitabashi et al., 2012) and several genera of *Ascomycota* (Muroi et al., 2016) under biodegradable mulch application. Cellulose and polyester degrading basidiomycetes are also common (De Souza, 2013; Watanabe et al., 2014). Even though nitrification processes reduced the ammonium concentration in microcosms after 16 weeks, the residual ammonium could have caused a repression of enzymatic function which can explain the reduced mulch degradation observed.

The presence of nitrifying bacteria in the soil microcosms was confirmed by increased *amoA* gene abundances in TN soils after 16 weeks and changes in soil ammonium and nitrate pools in both TN and WA soil microcosms. Bacterial *amoA* gene abundances varied between treatments in TN soils. However, nitrate concentrations increased, and ammonium decreased for all nitrogen treatments compared to no-nitrogen for both TN and WA soils. These observations point towards active nitrification processes taking place in the soil microcosms with added nitrogen. Furthermore, the reduced pH we observed with nitrogen amended soils could be explained by the formation of nitric





acid from urea and ammonium salts during nitrification (Yan et al., 1996). Previous studies have reported that urea-N additions can lead to significant increases in ammonia-oxidizing bacteria (AOB) abundance, activity and diversity (Chen et al., 2013; Hu et al., 2021a; Hu et al., 2021b).

There was minimal effect of plastic addition on the nitrifying potential of the soil microbial community. This suggests that the plastic material used in the study does not inhibit nitrifying populations to an appreciable extent. This

observation is supported by previous studies where use of Mater-Bi® demonstrated no ecotoxic effect on the soil (Ardisson et al., 2014) and BioAgri films did not significantly affect bulk soil quality or microbial communities (Bandopadhyay et al., 2020; Sintim et al., 2021; Sintim et al., 2019). The presence of plastic influenced nitrate concentrations only in TN soils where plastic addition reduced nitrate concentrations for all treatments. In both TN and WA soils, ammonium concentrations did not differ between plastic and no plastic treatments. Previous studies

have shown that plastic film mulches along with nitrogen fertilizers increased soil nitrate-N (Gao et al., 2009) and decreased or did not affect $N_2O$ emissions (Liu et al., 2014; Berger et al., 2013) under the films while on the surface. In this study, we saw that BDMs buried in soil had comparable effects to no-plastic treatments on nitrification processes.

### 4.3 Responses of soil hydrolytic enzyme activities to nitrogen and plastic amendments

A general trend of reduced soil enzyme activities was observed for all treatments after 16 weeks. Acidification of soils in the microcosms may have led to reduced enzyme activities as a result of effects on microbial physiology and base cation limitation (Shi et al., 2018). Reduced CB activity was observed for most nitrogen amendments after 16 weeks whereas a significantly increased activity was seen for the no-nitrogen control in TN. Reduced NAG activity has been reported in the literature post urea addition to soils at a rate of 392 kg N ha$^{-1}$ yr$^{-1}$ over a seven-year period (Zhang et

al., 2016). A few other studies have reported no change in BG activity after nitrogen application in temperate grasslands (Wang et al., 2014) whereas another study found that nitrogen addition over just a two-year period led to significant decreases in urease activity (Zhou et al., 2012). Decrease in peptidase activity has been documented by several other studies, where some report depressed activity after ten years of ammonium nitrate addition (Stursova et al., 2006), and others reporting dramatically reduced activities after nitrogen addition (Saiya-Cork et al., 2002). Thus,

these trends seem to point towards a general decline in enzyme activity after nitrogen addition in both short and long-term experiments. A negative relationship between NAG activities and soil $NH_4^+$-N content is also reported by Zhang et al. (2016) , indicating that ammonia could have an inhibitory effect on N-hydrolases. The only effect of plastic that was observed was for NAG activity in TN, where addition of plastic seemed to cause a greater reduction in NAG activity after 16 weeks compared to treatments without plastic. With field studies, it has been observed that addition

of BDM in soil has very little effect on the activities of common carbon and nitrogen cycling enzymes when compared to treatments without mulch, at least over a two-year period (Sintim et al., 2019; Bandopadhyay et al., 2020; Sintim et al., 2021).



**5 Conclusions**

In summary, we found that plastic decomposition was significantly reduced under inorganic nitrogen amendments, with the highest plastic decomposition seen for the no-nitrogen control, followed by amino acid treatments. Identical $CO_2$-C respiration trends were observed for both soils from TN and WA although percent biodegradation of plastic varied between locations. Evidence of increased soil nitrification was observed under nitrogen treatments compared

to no-nitrogen. Reduced soil enzyme activity rates after 16 weeks suggested a potential inhibitory effect of nitrogen activity on hydrolases. The effect of plastic addition had minimal effects on nitrification processes and enzyme activities after 16 weeks. The results indicate that addition of nitrogen fertilizers in cropping systems with tilled-in BDM fragments may inhibit decomposition of mulch material without compromising nitrification processes.

**Data Availability** All data used in the preparation of this paper are available at the repository figshare (doi: 10.6084/m9.figshare.21566019)

**Author contributions** Conceived and designed the experiments: SB JMD DGH SMS. Performed the experiments: SB, ME, MBA, MS, JH and JLG. Analyzed and interpreted the data: SB, ME, MBA, JMD, DGH. Wrote the paper:
SB and JMD with inputs from other authors.

The authors declare that they have no conflict of interest.

**Acknowledgements**
We thank Dr. Sindhu Jagadamma and Shikha Singh for the training and help provided to operate the Shimadzu Gas
Chromatograph, Xiangru Xu and Tori Beard for help with soil processing and Dr. Arnold Saxton for statistical advice. This work was funded USDA grant 2014-51181-22382 to DGH, SMS, and JMD.




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
