# Peer review of "Organic and inorganic nitrogen amendments suppress decomposition of biodegradable plastic mulch films"

_EGUsphere, 2022_

## Author Response (AR1)

*Thank you for the opportunity to revise our manuscript. The reviewers provided many helpful comments and we feel our revised manuscript is strengthened as a result. Please find our point by point responses in >blue bold text below.*

**RC1**

This is an interesting study assessing the effect of N fertilization of soils on biodegradation of biodegradable polymers and starch. The study is carefully conducted and designed and the statical analyses seems stringent and correct.

There are, however, a number of points the authors may want to consider addressing to improve the clarity and constrain the implications to what can be inferred from the experiments. These points are part of the specific comments made below. To highlight a few:

- The study set out to test the effect of fertilization (N) on biodegradation. It seems that this may warrant the use of soils that are N deprived (or span a gradient in N availability). Yet, the two soils chosen here have relatively high N contents to start. How generalizable are results then?

  **>The authors thank the reviewer for pointing this out. This is a valid point. Our study aimed to understand the effects of added nitrogen to field-relevant soil. We intentionally did not alter the chemistry of the soil so we could get closer-to-field condition responses. We are happy to add a few sentences in the last paragraph of the introduction to note this in lines 108-112.**

- The plastics contain starch and PBAT. Starch seems to degrade preferentially. So are we observing an effect of N on starch? Maybe the effect on PBAT is different?

  **>Starch degrades preferentially from our results. While we did not explicitly test the effect of N on starch and PBAT, we can only assess this on a relative basis based on thermal degradation process. We show the relative composition of starch and PBAT (supplemental figure S5) after incubation, and note that there was a reduction in % starch component for some of the nitrogen amendments compared to the field-weathered control mulch.**

- The duration of the experiment is "only" 16 weeks, while biodegradation in standard tests is followed over 2 years. Is it valid to infer that what one sees initially (during 4 of 24 months) is applicable to the remaining 20 months?

  **>This is a valid point, and we are happy to address this in the discussion in lines 638-640. Our focus was on acute or early effects of a nitrogen**

**fertilization event on degradation; therefore would not necessarily predict degradation over longer time scales (years). Correlating this early reduction in degradation to performance on standard 2 year biodegradability tests would be a valuable follow up experiment for the future.**

- The effect of N addition on the efficiency at which carbon is incorporated into microbial biomass is not addressed. Could it be that less CO2 formed because microbes build more biomass in a world of excess N?

  **>This is a valid and interesting point, and we are happy to address this in the discussion (lines 641-645). We did not measure microbial biomass, but yes, we know that changing nutrient conditions can certainly change microbial carbon use efficiency, and typically that a decrease in C:N leads to an increase in CUE. So we can't rule out that some of the CO2 decrease was due to changes in biomass. However, we see that our CO2 data corresponds well to the degradation seen in the material images taken at the end of the experiment, suggesting that CO2 was a reliable indicator of degradation of the mulch in our experiment, and we can conclude that biomass changes would have only minimally contributed to the observed change in CO2.**

  It seems that N addition affected also general soil activity, microbial abundance and enzyme activities. If this is true, then the reviewer suggests rethinking the title. The tile suggests that the effect is on plastic "only". But one could argue: the soil simpl did not do well after N addition and, no wonder, did this also affect plastic biodegradation

  **>We thank the reviewer for raising this point. While it is true in certain cases, this was not observed across all treatments for microbial abundance and enzyme activities. In certain cases, we see less of an inhibition in these metrics in nitrogen treatments compared to no-nitrogen treatments post incubation. For example, there were minimal to no statistical differences in bacterial gene copy changes across treatments after 16 weeks. However, given that we do see an effect of N in certain cases, particularly enzyme activity, we are happy to include this in the discussion (lines 694-698). For the title, we now use the word "reduce" instead of "suppress" to avoid misinterpreting this as an irreversible process.**

More specific comments and more elaboration on these points are given below.

**Specific comments**

Line 13: microcosms: unclear what is meant (laboratory incubations?)

**>By microcosms, we are referring to the experimental units, in this case mason jars where the soils were incubated for 16 weeks. We have replaced this term to "laboratory incubations" to make it more clear in line 14.**

Line 19: the authors refer to "decomposition" because mineralization to CO2 was not followed?

**>We would replace "decomposition" with biodegradation throughout the text for consistency because CO2 was measured.**

Line 20: Now authors refer to "biodegradation". Strictly speaking this is possibly only based on respirometric tests or if residual polymer is quantified.

**>We use the term "biodegradation" because in this experiment, we measure CO2 as a respirometric test to assess biodegradation. While we do not quantify residual polymer, we provide visual images of residual polymer taken at the end of the incubation.**

Line 23: suppression by 1%. That seems to be little. One % can be detected also given the variations between incubations?

**>This is a valid point, and we are happy to mention this caveat in the discussion in lines 565-567, especially where there are minimal differences in mulch degradation between nitrogen and no-nitrogen treatments.**

Line 25: Are the nutrients themselves slowing down decomposition or does nutrient addition change other soil parameters that then affect degradation

**>While nutrient additions change soil parameters in certain treatments, this was not observed across the board. However, given that there were some changes, we are happy to address this in the discussion and mention that the effect we observed may have been direct or indirect. We address this in lines 693-698.**

Line 50: why is a "composting" standard cited when biodegradation is supposed to happen in the soil?

**>We cite this to make the readers aware that this is the most widely cited standard in reference to BDMs. In the next line we note the release of EN17033 standard, which was the first standard to certify plastic mulch films as "biodegradable" in ambient soil.**

Line 54: not 90% biodegradation but rather 90% conversion to CO2. That is a difference because if some of the polymer carbon is used for biomass formation, mineralization to 90% may underestimate biodegradation

>The authors thank the reviewer for pointing this important difference. We have now edited these lines to reflect that "biodegradation" here is measured via $CO_2$ production and hence is, more appropriately, mineralization. We have changed this in lines 56-57.

Line 75: shift in which direction (maybe say: C:N is lowered)

>We have made this change in the manuscript in line 77, as it is more specific.

Line 83: "a few studies" … and which explanation was provided? Are results really mixed or do most studies show a positive effect of fertilization on organic matter decomposition?

>Most studies do show a positive effect of N addition on organic matter decomposition. We cite the literature for this in lines 80-83. For the few studies that show a negative effect, we include supportive literature in lines 83-85 suggesting that responses can be in either direction depending on field conditions and management history.

Line 88: proposed … may .. – maybe better "… speculated that …"

>We are happy to accept this change and incorporate it in line 90.

Line 95: If the authors say "biodegradation", then they need to conduct respirometric measurements. Otherwise, if weight loss (or similar) is measured, its strictly speaking not biodegradation but rather decomposition or dissipation

>For this experiment, we conduct respirometric tests ($CO_2$ evolution) to assess biodegradation and report those values. Hence, we use the term biodegradation. We did not measure weight loss of mulch but do provide detailed mulch imaging after the end of incubation.

Table 1: C/N ratios are relatively small (around 10). Would one expect to see nitrogen limitation in such soils in the first place?

>This is a valid point. It is possible that these soils were not nitrogen limited to begin with and we have elaborated on this in the discussion (line 570-572). Given that we wanted to conduct these assessments on field relevant soil, we did not manipulate N content of the soils. Instead, we chose field soils that were actively being treated with mulch to also have the legacy of the microbial community to mulch exposure. We have added a note to elaborate on this decision in lines 108-112.

Line 130: how could the authors be sure that the sole mass was mulch film and not including the mass of some adhering particles, etc.

>Because we used field-weathered mulch, we can't entirely exclude some soil contamination. Mulches were cleaned and brushed to remove as much of the adhering particles as possible, but of course there may be some remaining. However, since we exposed the whole roll of mulch to the same conditions in the field, and all mulches were then similarly cleaned/brushed to remove particles, we assume that any adhering particles would be consistently present throughout the mulch and across treatments.

Line 140: in some of the treatments, not only N but also labile C was added. How can the authors delineate the two? Sure, the organic N was added as an N source. But given that it is also a C source, is it not possible that labile C addition (in combination with N) suppressed BDM decomposition because there was no need to access its carbon? What is the ratio of BDMF added C versus fertilizer added C? Also, how far did fertilizer addition change the soil C:N?

>This is an excellent question. Carbon was present in two treatments, the amino acid treatment (which was added as a complete supplemental media) and urea. For the amino acid treatment, we observed relatively less suppression of mulch decomposition compared to the other nitrogen treatments. In urea, we did observed more reduction in comparison to the other nitrogen treatments, so it is possible for urea treatment that the C addition was relieving the need to access mulch C. However, we note that the ratio of mulch C to amendment C was roughly 7.65 for the amino acid treatment and 56.55 for urea. Given that the urea C input was particularly low, we anticipate its contribution to be minimal. We have added a paragraph to address this excellent point (lines 629-636).

We noted the changes in soil C:N in response to the various N additions in Table S3. There were minimal differences in C:N between treatments within TN and WA. However, when doing three-way ANOVAs, including both locations, we report a significant difference among nitrogen treatments in C:N ratios (Table S2).

Line 152: Do 16 weeks suffice if regulatory standards allow for 2 years?

>Please see our response to the general comment above regarding the time frame.

Line 165; "rate equivalent". Unclear what is meant. The authors always added the same amount of N?

>Rate equivalent means same concentration. We rewrote this sentence to clarify (line 180).

Line 185: It seems that the mesocosms were set up to determine CO2 formation. The reviewer assumes that biodegradation was quantified from the excess CO2 formation in

incubation vials with soil + BDMF relative to only soil (no BDMF) incubations. If correct, then the controls are subtracted from the incubations to arrive at polymer-derived CO2 formation.

However, what about the controls for N amendment? Were separate control incubations run for all N amendments? Because N amendment can change background mineralization (and organic N addition will for sure produce extra Co2)

>Yes, we have included all these controls in our experiment. We have controls for N amendment which are the no-nitrogen treatments. The no-nitrogen treatments were established both with and without mulch for cross comparison across all treatments. The experimental design can be found in line 148-152.

Line 213: from "soil samples". How close where these to the foil pieces or which soil mass was extracted? Quite clearly, if there is an effect of foils on soil microorganisms, it should be higher close to the foils. This means that a very reproducible soil sampling must have been done?

>We collected soils by sieving the soil and mulch mixture from each jar post incubation. During the incubation set up, mulches were buried within the soil and these methods are in lines 163-165. Given that this was a fairly small volume packed with soil and mulch we hypothesize that there was quite good contact between soil and mulch. We elaborate on this in lines 165-167.

Line 235: was quenching by soil DOM accounted for? Were calibration curves of fluorescence standards prepared in soil extracts or pure buffer solutions?

>All calibration curves of fluorescent standards were prepared in soil extracts such that each soil sample had its own calibration curves. We have added this information in lines 251-252 to make it clear.

Line 260: did the visual analysis confirm/agree with the mineralization data?

>Yes, we did see comparable results between the visual analysis and the CO2 data. We show these data in Fig 2, S1-S4 and address this in lines 386-390.

Line 274: The question above on the controls for soil + polymer + fertilizer refers to this equation. Essentially, were there no mulch + fertilizer controls?

>Yes, for each fertilizer treatment we have + and – mulch levels to calculate % biodegradation and we show this separately in a table in Supplemental Table S4.

Line 289: changes in the chemical bonding... which bonds are referred to? Polymer bonds?

>Yes, we do mean polymer bonds here. The description of these bonds can be found in Supplemental Table S7 which we have referenced in this line 304 to make it clear.

Line 314: what are "main effects" and "interaction effects"?

>The main effect referred here is the effect of "location" with levels: TN and WA. Interaction effects include the interaction terms of location with plastic and nitrogen in our model. We have clarified this in lines 329-330.

Line 325: so could the nitrogen effect be, in fact, a soil pH effect?

>It is possible that the nitrogen could impact pH and that would impact degradation. But we consider this effect as ad-hoc since urea and nitrate would be applied as fertilizers in the field which would impact pH and possibly mulch biodegradation. Our experiment does not necessarily indicate whether the effect of fertilization is direct or indirect (e.g. via pH modification), and we have included a comment about this in the discussion in lines 693-697 and 583-586.

Figure 1: soils are quite acidic (pH 5.25). is this normal for these soils? Are the soils not carbonated?

>The starting soil pH is referred in Table 1. Both Knoxville and Mt Vernon soils have pH close to 6, and this is well within the normal range for these soils. After 16 weeks, we see that in WA soils the pH reduced by 0.5 unit whereas for TN soils it reduced by 0.25 unit; this is still within normal ranges.

Line 365: Did the authors assess which components of the films degraded? And if nitrogen addition affected biodegradation of one component more than others? Also, biodegradation is inferred from Co2 formation. Is it possibly that microbes incorporated more C into their biomass in the presence of N, thus leading to an "apparent" decrease in biodegradation only?

>The main component of mulch that showed preferential degradation is starch. This was indicated by the thermal degradation data (TGA). Nitrogen amendments, specifically amino acid and ammonium nitrate had significantly reduced starch content compared to field-weathered controls (Fig S5 and lines 410-413).

>Please see our response above to the general comment about biomass changes.

Line 370 time and t being proportional: remove, since this indeed is the programmed heating rate

>We have removed this phrase (line 393).

Line 378: and possibly more efficient biomass incorporation?

**>While we do not have biomass data to provide, we have noted this important point in the revised manuscript (line 398).**

Line 382: "must occur"? unclear what is meant.

**>We have changed it to "during biodegradation" (line 403).**

Line 389: how was this calibrated? Simply relative peak areas? Do the authors know if responses scale linearly with mass?

**>These data were based on relative peak areas. As it is relative data, it cannot be correlated to absolute mass.**

Figure 2: what are these numbers relative to C added (mineralization in % of carbon added).

**>The numbers relative to C added is the % biodegradation values that we report in the paper. This is based off the methods in Methods section 2.7.2 and the data for this is in the Supplemental Table S4.**

Figure 2a: very hard to read. Maybe pull apart in separate graphs (multipanel)

**>We have moved Fig 2a into the supplemental (now Figure S1) and kept Fig 2b in the main text with added statistics (now Figure 2).**

Figure 2: it seems that N additions not only affected polymer mineralization but also soil background mineralization. Did the authors check for that? This is important because then the conclusion shifts from "N addition slows down biodegradation of plastics" to "N addition slows down soil carbon respiration to CO2, including respiration of added plastic carbon".

**>This is a fair point. However, we do note that statistically there were minimal differences between treatments without mulch (including all the nitrogen and no-nitrogen treatments). This is shown in Fig 2. Also, the biodegradation curve in Fig S1 shows that these treatments are much less different from each other than with added plastics. We have made a note about the slight reduction in CO2 respiration due to N addition in lines 383-384.**

Line 457: see comment above on effect of N on bacterial abundance and/or activity versus effect on film degradation.

**>Thanks – response above!**

Line 527: It is unclear whether the 10% comes from starch or from PBAT?

**>This is a valid point. We are unable to directly assess if the 10% biodegradation is through use of starch or PBAT. While our study indicates that starch is preferentially utilized, we cannot directly tie this in with the % biodegradation. We now mention in the discussion that use of isotopically labelled mulch C can aid to answer this more accurately (lines 613-620).**

Line 528: It should be mineralization not decomposition as biomass uptake is not determined (nor residual polymer quantified)

**>Good point; we have replaced "decomposition" with "mineralization" (line 546).**

Line 531: since this study is on the "early stage" of biodegradation, how can the authors rule out that N addition has positive effects in later stages? Should this "early stage" fact not be stated more prominently as to be clear that the study was relatively short?

**>This is a fair point. It is possible that longer exposure could change and possibly increase biodegradation. This study was a short incubation compared to the two years typically used to assess biodegradation. Please see also our response to the general comment above.**

Line 353: if there is discrepancy, what does this tell us about using the photographic method to "quantify" biodegradation in field studies. Is this a reliable approach?

**>While there are constraints to this approach, we still argue that it gives good relative estimates across treatments. Visual estimates cannot be used as the sole determinant for biodegradation and is typically more of a supportive data since it is less quantitative. Methodically, it is challenging to recover mulches which have degraded to fine particles using a sieving method and better approaches need to be developed for recovery of mulch particles and accurate assessments. We have made a note of this in the discussion in lines 555-559.**

Line 539: but also of background soil organic matter decomposition?

**>Correct, we do see minor suppression of background organic matter decomposition as well, so we have added a note in this section regarding this point in line 561-563.**

Line 540: you mean by 1 percentage point (from 4 to 3%) or, as stated, by 1% (from 4% to 3.96%). Same for the 6%

**>We have clarified this by saying 1 and 6 percentage points in lines 563-564.**

Line 544: indeed, C:N was 10!

>This is a fair point; please see our response to the general comment above about our intent to use of field soil with history of mulch exposure for more field-relevant assessments. We intentionally did not alter soil chemistry for these soils.

Line 547: see comment above. Is the conclusion then not rather: N addition lowers microbial abundance/activity and thereby also biodegradation of polymer? That conclusion is very different than how the title of the manuscript reads right now.

>We have acknowledged this in the discussion of the manuscript as a possible reason for reduced mulch degradation (lines 694-698). For simplicity, we keep the title as before but change the word "suppress" to "reduce" to avoid misinterpreting this as an irreversible process. In these lines we have added a point about the fact that we are not making claims of direct vs. indirect mechanisms. We note that not all metrics of microbial activity were reduced: e.g. nitrification potential remains intact in these soils post N amendment, which is often used as a metric for soil health. Bacterial gene abundances also remain unaltered post N amendment (Fig S6A). However, most enzyme activities show a decline, so we acknowledge effects on soil activity.

Line 558: in this case its also not the N that does harm but rather the pH

>We have made the point about pH clear in this section in lines 584-586.

Line 580: how could priming be avoided?

>In our case, we add carbon in the form of mulch, so it was not avoidable. However, mulch C inputs to soil in the field is typically mulch lower than other forms of organic matter that supports priming processes. So, we anticipate this process to be minimal in our system and added a comment about priming (lines 618-621).

Line 583: what is meant by "labelled"? Any example?

>By labelled substrates/ mulches we refer to isotopic carbon signal from mulches. For example, use of 13C labeled mulch to track $CO_2$ production specifically from mulch or components of mulch. We have clarified this in line 612.

Line 585: has this been done already?

>Yes, there are studies that have used such a process such as a novel approach by Zumstein et al. 2018. We have included relevant literature to this effect (line 618).

Line 588: led to is better than "caused". Because N addition unlikely was the cause for depolymerization

>**Good point - we have changed the wording as suggested (line 623).**

Line 630: see comment above on what the effect was: an effect on biodegradation of polymers or a general effect on soil activity. Also, if one wanted to test for N limitation (and a potential benefit of adding N), would one not chose soils with varying C:N ratios? Including soils that are N deprived?

>**Thank you; please see our comment above.**

**RC2**

This article describes a research based on an articulated, well-constructed and well-analysed experimental scheme. This article has the merit of highlighting how some prejudices can condition scientific thought and that every statement, even the most obvious, must be verified. The statement "The addition of nitrogen improves the microbial metabolism and thus also the biodegradation capacity" is examined here and surprisingly reversed.

In my opinion, this work is well made and definitely deserves to be published. Below I report some observations that mainly concern editorial or secondary issues or some statements that require some further discussion.

Title: The term "*suppress*" seems to indicate an irreversible block of decomposition. However, the highlighted effect is a reduction in the rate of biodegradation rather than a suppression. Furthermore it is not known whether, once the inhibition due to the excess of nitrogen has been removed, i.e. after its absorption in biomass, etc. this inhibition disappears or not. It is not known whether this is a transient or irreversible effect.

**This is a valid point, and we are happy to change the title to reflect it. We now say "reduce" instead of suppress.**

Line 25 "*This study suggests that addition of nitrogen, particularly inorganic amendments, could negatively affect mulch decomposition but that mulch decomposition does not negatively affect soil nitrification activity.*" The correct term is "*slow down*" rather than "*negatively affect*" mulch decomposition. From a practical point of view, it could be a positive effect. In fact, fertilization takes place at the beginning of the agronomic cycle, i.e. when the mulch has to resist. At the end of the cycle, the crop has absorbed nitrogen and therefore the greatest potential for degradation occurs just when the mulch has finished its function.

**This is an excellent point and we thank you for pointing this out - we have now changed the phrase "negatively affect" to "slow down" in line 27.**

Line 56 "*Climate, crop, weathering and soil management can all influence biodegradation rates and complete biodegradation is not always achieved (Anunciado et al., 2021; Hayes, 2018).*" Environmental factors can slow down the achievement of complete biodegradation, which however will occur if the material is fully biodegradable. Maybe this can be rephrased as following: Climate, crop, weathering and soil management can all influence biodegradation rates and complete biodegradation is not always **fast** achieved.

**This is a good suggestion, so we rephrased this sentence (line 60). The reviewer is correct although we point out that complete biodegradation is sometimes not achieved due to additional components within commercial mulches such as additives and plasticizers.**

Line 128 Field-weathered mulch samples are used in the experiment. While I understand the reason for this choice (that is, to introduce the aging factor in the experiment), the use of material recovered from the field poses the risk of using a heterogeneous test material. Perhaps a brief discussion about this risk is needed.

**We are happy to add a few sentences highlighting the possibility of the field-weathered mulch being a heterogenous material with altered chemistry and adherent soil particles and microbes. We know from our previous work that field-weathered mulch films degrade differently that "off the roll" fresh mulch films, and intended our experiment to more closely approximate field processes, even if that means introducing more heterogeneity and variability to the system. We added a discussion of this point (lines 139-144).**

Line 146. It is reported that the added nitrogen content is similar to that applied in agronomic reality. It would be helpful to elaborate this point in the discussion.

**We are happy to add a few sentences to this effect in the discussion in lines 588-591.**

Line 150. It is difficult to understand how the plastic pieces were placed. Can you improve this text?

**The plastic pieces were layered within the soil such that they were entirely covered. We have added some more detail in lines 163-165.**

Line 197, A typo : % gravimetric soil moisture

**Thank you, we have corrected this in line 210.**

Line 268.  It is more correct to say: "..*the extent of aerobic biodegradation of BioAgri*..."

**We have changed this wording in the revised manuscript in line 285.**

Line 274. Equation 3 indicates the % biodegradation, not the *theoretical* biodegradation. The term theoretical biodegradation is generally used to indicate the total amount of $CO_2$ that can be produced by a material in the event of total oxidation of the carbon it contains.

**We have corrected this to simply % biodegradation in line 289 and 291.**

Line 320. The description of the results, although correct, is sometimes difficult to follow and causes difficulty in reading, which is solved by going directly to the Figures. In fact, the text begins by describing the small differences (perhaps *statistically* significant but not significant in absolute terms) and only later are the most relevant changes described. In this case, for example, it would be simpler to say something like this. The addition of nitrogen, in any form, causes a significant increase in the EC both in the presence and in the absence of plastic. In fact, no or small, although sometimes statistically significant, reductions in comparison with blank soil is due to plastic. Likewise (for example): the addition of nitrogen, in any form, causes a significant drop in pH both in the presence and in the absence of plastic, with the exception of urea where the pH drop is neutralise by plastic.

**This is a great suggestion, and we have reorganized the writing in lines 338-341 as suggested.**

Fig. 2a It is very difficult to read the legend. Maybe you can split Fig 2 into two separate figures and enlarge legend of Fig a and use larger symbols.

**We have moved figure 2A to supplemental (Figure S1) and enlarged the legend. The orginial Figure 2B is now Figure 2 in the main manuscript.**

496 there is a typo in the legend: standard error.addition

**Thank you – fixed (line 513).**

Fig.6 (b) Some of the bars indicating $NO_3$ concentration show "amoA" and an arrow. Can you please clarify this in the legend.

**We have added an explanation to figure legend (line 515-16).**

Line 527. You report a 10% and 4% biodegradation. However, no biodegradation curve is reported. Maybe a biodegradation curve at least in Supplementary information could be of help.

**We have provided biodegradation curves by representing CO2-C evolved over time in the original Figure 2a, which we have now moved to the supplemental (Fig S1) at the request of both reviewers. In the supplemental information we have further provided biodegradation percentages based on the starting C content of the mulch (Supplemental Table S4). This was calculated by subtracting the CO2-C evolved in the +mulch and -mulch treatments dividing by the starting C content of the mulch. We report the methods in Methods section 2.7.2.**

Line 531, What exactly is "*the anticipated carbon source for the early stage of biodegradation*"? Can you elaborate on that?

**We suggest that starch is the anticipated source for the early stage of biodegradation. We have clarified this to make it clear in line 551.**

Line 540. Are the reported values (6% and 1%) percentage points or the percentage reduction? I assume percentage point but not having a biodegradation curve available it is difficult to fully understand.

**These values are percentage points and we have reworded this to clarify (line 563-564).**

---

## Author Response (AR2)

Dear authors,

I have to apologize for the slow response due to some difficulties in getting a re-review of your manuscript. However, I have now again in detail reviewed the revised version of your manuscript and your responses to the reviewer comments and am pleased to inform you that I recommend the manuscript for publication with minor technical corrections (see note to authors).

Thank you once again for sending your scientific work to SOIL.

With best regards

Peter Fiener

Additional private note (visible to authors and reviewers only):

Dear authors,

in the following, some specific comments you should account for in the final version of the manuscript.

Line 32-33: I think it would be fair to add 'slightly reduced BMD degradation' as the overall effect is not that substantial.

**>Changed**

Line 152: L-arginine should be in 'mg' and not 'mg/L'.

**>corrected**

Line 285-286: I am wondering that you give 1309.41 µg and not just 1309 µg?

**>Changed to 1309**

Line 323: Give some details how you removed the outliers.

**>Added**

Line 363: In Table S3 you give the reduction in C:N values but not the reduction in %C and %N.

**>We added the %C and %N data for the initial samples to Table S3 so that this reduction can be seen.**

Figure 6b and 6c are very small and labels hard to read.

**>We changed the layout of this figure so the panels are all stacked vertically, which allowed us to make b and c larger and more readable.**

Line 555: "… while there are constraints …" a reference would be good.

**>added two references to support this argument**

Line 566: "… here that the 1% reduction in mulch…" should be "…here that the 1 percent point reduction in mulch …"

**>corrected**

Line 569: replace "repression" by "reduction".

**>replaced**

Line 705: As the differences in biodegradation are small (but significant), I suggest making this also clear in the conclusion, e.g. "… plastic biodegradation was slightly but significantly reduced …"

**>changed to suggested wording**

Figure S1: The caption of the figure does not fit to the figure (e.g. bars, standard error, lowercase letters etc. are not shown in the figure.
**>corrected**